# 15-Year Analysis of Direct Effects of Total and Dust Aerosols in Solar Radiation/Energy over the Mediterranean Basin

Kyriakoula Papachristopoulou [1,2,3,*], Ilias Fountoulakis [2], Antonis Gkikas [2], Panagiotis G. Kosmopoulos [4], Panagiotis T. Nastos [1], Maria Hatzaki [1] and Stelios Kazadzis [3,4]

1   Laboratory of Climatology and Atmospheric Environment, Sector of Geography and Climatology, Department of Geology and Environment, National and Kapodistrian University of Athens (LACAE/NKUA), 15784 Athens, Greece; nastos@geol.uoa.gr (P.T.N.); marhat@geol.uoa.gr (M.H.)
2   Institute for Astronomy, Astrophysics, Space Applications and Remote Sensing, National Observatory of Athens (IAASARS/NOA), 15236 Athens, Greece; i.fountoulakis@noa.gr (I.F.); agkikas@noa.gr (A.G.)
3   Physicalisch-Meteorologisches Observatorium Davos, World Radiation Center, 7260 Davos, Switzerland; stelios.kazadzis@pmodwrc.ch
4   Institute for Environmental Research and Sustainable Development, National Observatory of Athens (IERSD/NOA), 15236 Athens, Greece; pkosmo@noa.gr
*   Correspondence: kpapachr@noa.gr

**Abstract:** The direct radiative effects of atmospheric aerosols are essential for climate, as well as for other societal areas, such as the energy sector. The goal of the present study is to exploit the newly developed ModIs Dust AeroSol (MIDAS) dataset for quantifying the direct effects on the downwelling surface solar irradiance (DSSI), induced by the total and dust aerosol amounts, under clear-sky conditions and the associated impacts on solar energy for the broader Mediterranean Basin, over the period 2003–2017. Aerosol optical depth (AOD) and dust optical depth (DOD) derived by the MIDAS dataset, along with additional aerosol and dust optical properties and atmospheric variables, were used as inputs to radiative transfer modeling to simulate DSSI components. A 15-year climatology of AOD, DOD and clear-sky global horizontal irradiation (GHI) and direct normal irradiation (DNI) was derived. The spatial and temporal variability of the aerosol and dust effects on the different DSSI components was assessed. Aerosol attenuation of annual GHI and DNI were 1–13% and 5–47%, respectively. Over North Africa and the Middle East, attenuation by dust was found to contribute 45–90% to the overall attenuation by aerosols. The GHI and DNI attenuation during extreme dust episodes reached 12% and 44%, respectively, over particular areas. After 2008, attenuation of DSSI by aerosols became weaker mainly because of changes in the amount of dust. Sensitivity analysis using different AOD/DOD inputs from Copernicus Atmosphere Monitoring Service (CAMS) reanalysis dataset revealed that using CAMS products leads to underestimation of the aerosol and dust radiative effects compared to MIDAS, mainly because the former underestimates DOD.

**Keywords:** aerosols; dust; direct radiative effects; solar energy; Mediterranean Basin

## 1. Introduction

Aerosols modulate the radiation field of Earth's atmospheric system with several implications for life on earth. The physical mechanisms through which they influence the radiation budget are manifold. Aerosols interact directly (direct effects) with the shortwave (SW) and longwave (LW) radiation through scattering and absorption. Moreover, through their interactions with clouds, they have semidirect and indirect effects by altering the atmospheric conditions related to cloud formation/dissipation due to absorbing aerosols and by acting as cloud condensation and ice nuclei (altering the microphysical and hence the optical properties of clouds) (e.g., [1]). The direct radiative effects of aerosols refer to changes in radiative fluxes due to the direct aerosol–radiation interaction (radiative effects due to aerosol–radiation interactions (REari), as renamed in the IPCC AR5 [2]).

Cloud-free aerosol direct radiative effects depend on aerosol optical properties such as the spectrally resolved aerosol extinction coefficient or the integrated aerosol optical depth (AOD), the single scattering albedo (SSA), the scattering phase function or the integrated asymmetry parameter and other environmental parameters such as surface reflectance and the concentration of atmospheric trace gases [3–7]. Their magnitude corresponds to the perturbation of the radiation fields induced by aerosols, within the Earth's atmospheric system, with respect to an atmospheric state without their presence.

Aerosols are among the main climate change drivers, and estimates of changes in their radiative effects are highly significant for climate-change-related policy making as they are linked to changes in surface temperature [8]. Nevertheless, estimates of the total anthropogenic radiative forcing are still highly uncertain [9]. Several reasons contribute to these not yet well-constrained estimates, such as the heterogeneity of the processes governing aerosols' production and removal, which in turn regulate optical and microphysical properties, both determining the associated aerosol–radiation interactions. Focusing on Earth's surface, apart from the importance of aerosols on climate, studies quantifying the impact of total aerosols (natural and anthropogenic) on incoming solar irradiance are also essential for other societal benefit areas such as the energy sector (e.g., [10–12]).

Dust particles constitute a major component of atmospheric aerosol load [13,14]. Their emission is primarily wind-driven over the arid or semiarid regions of the planet. The Sahara Desert in North Africa, the Arabian and the Asian deserts are the major dust sources on Earth, with the highest contribution to global dust load (more than 50%) being emitted from North Africa [15–17]. Even though the dust sources are localized, the spatial distribution of dust over the globe is extensive due to dust mobilization under favorable meteorological conditions. Dust can be transported over long distances with significant implications for the climate of the affected areas and the global climate [18–20]. In addition to its significant radiative effects, dust also plays a key role in other processes such as the productivity of oceanic waters [21] and terrestrial ecosystems [21] and affects human health [22–24]. The diameter of dust particles is of the order of 0.5–50 μm, and thus the Ångström exponent of dust aerosols is smaller than 1 [25]. Dust particles generally scatter and redistribute rather than absorb solar radiation (SSA > 0.9 at visible wavelengths), although dust absorbs solar radiation at short (i.e., ultraviolet) and long (i.e., infrared) wavelengths more effectively [26–28].

The Mediterranean Basin, located at the crossroads of air masses from all over the globe [29], experiences high aerosol concentrations of both natural and anthropogenic origin, and the aerosol spatiotemporal variability over the area has been investigated in several studies (e.g., [30–33]). In addition, due to its proximity to the Earth's most active dust sources located across the Sahara Desert and Middle East deserts, the Mediterranean Basin often experiences high dust aerosol loads (dust intrusions). Several studies have investigated the Saharan dust transport towards the Mediterranean Basin by exploiting, either solely or combined, in situ measurements, remote sensing (ground-based or satellite) retrievals and atmospheric dust numerical products (e.g., [34,35]). Across the Mediterranean Sea, dust concentrations fade down from south to north because of the removal of mineral particles from the atmosphere, either due to dry or to wet deposition [36]. It should be noted that long-range transport of Saharan mineral particles has also been recorded over northern Europe [37,38]. Additionally, there is a distinct seasonal cycle of the longitudinal spatial distribution of dust with higher concentrations in spring at the eastern and central Mediterranean and in summer at the western Mediterranean [36,39,40]. The seasonal patterns are related to the seasonality of the prevailing meteorological conditions over the area [36,41]. This intricate aerosol regime makes Mediterranean Basin one of the most interesting areas for investigating the aerosol and dust radiative effects.

Towards the reduction of greenhouse gas emissions and climate change mitigation, the deployment of renewable energy technologies is one of the main pillars [42], while at the same time renewable energy technologies provide energy in a sustainable manner. Among renewables, the share of solar technologies is growing fast. For electricity production, the

systems in use are the photovoltaic (PV) cells and the concentrating solar power (CSP) plants, which harness different components of the solar irradiance: the global horizontal irradiance and the direct normal irradiance, respectively [43,44]. Within the North Africa, Middle East and Europe (NAMEE) domain, International Energy Agency (IEA) projections up to 2025 indicate a continuous growth in solar PV capacity, while some countries (e.g., Morocco) are increasing their CSP capacity [45].

Over most of the globe, the availability of downwelling surface solar irradiance (DSSI) depends mainly on attenuation by clouds, but the role of aerosols is also significant and under certain conditions can be dominant (e.g., [46,47]). The countries around the Mediterranean Basin, apart from their prolonged sunshine durations, also experience high levels of aerosol loads, mainly composed of mineral particles. In the recent study by Fountoulakis et al. [47], it was found that for Cyprus in summer the GHI attenuation by aerosol is comparable to the attenuation by clouds, while solely dust attenuates more DNI than clouds. In the same study, it was found that 30–50% of the overall DSSI attenuation by aerosols was due to dust. For lower latitudes with even more rare cloudy conditions, aerosols and specifically desert dust constitute the most common source of DSSI attenuation, as was demonstrated in a study for Egypt by Kosmopoulos et al. [10]. It must be pointed out here that even for southern Europe, dust can substantially reduce surface solar radiation during dust intrusions [48]. Thus, studying the radiative effects of dust is of great interest for the broader Mediterranean Basin. Existing studies are limited over specific locations or dust event days (e.g., [10,47,48]).

In order to derive the aerosol radiative effects, continuous monitoring of their optical properties is essential [49]. AOD is a quantitative measure of the integrated aerosol extinction in the atmospheric column; hence, it constitutes the most important aerosol optical property for estimating the aerosol direct radiative effects [50]. In the same manner, DOD is a proxy of the mineral dust particles' load, in optical terms, throughout the atmospheric column. The most accurate estimates of those optical properties are provided from ground-based remote sensing instruments (sun photometers), which are however sparse, and their geographical distribution is spatially inhomogeneous. This observational gap is complemented by satellite remote sensing along with the related retrieval algorithms, providing considerably accurate measurements with global coverage and high spatial and temporal resolution. Although they are still not as accurate as ground-based estimates of the aerosol optical properties, the accuracy of satellite-based aerosol products has been improved substantially in recent years [51,52]. A significant disadvantage of satellite products is that they are representative of relatively wide areas rather than specific locations, and thus they can be highly uncertain when they are used to study aerosols over complex environments [53,54]. MODIS instrument onboard Aqua (since May 2002) and Terra (since December 1999) satellites provide AOD and other valuable information for aerosols worldwide. Based on MODIS AOD product, a new DOD dataset was generated, the ModIs Dust Aerosol (MIDAS) dataset [55], enhancing the existing dust aerosol monitoring capabilities. Considering its global coverage at fine spatial and temporal resolution, over a long period (2003–2017), this dataset is suitable for dust climatological studies (e.g., [47,56,57]).

Another tool for dust monitoring, on a regular basis and for long time periods, relies on atmospheric models simulating aerosols' life cycle. Via the assimilation of quality-assured observations, quite reliable numerical products are also available from reanalysis datasets [58–60]. For aerosol-related studies, one of the most exploited reanalysis datasets is the Copernicus Atmosphere Monitoring Service (CAMS) dataset [61] providing aerosol optical properties for the total load, as well as for aerosol species. However, compared to satellite products, its spatial resolution is relatively low, and regardless of the assimilation process of aerosol observations, the performance of their outputs depends on modeling aspects (e.g., the balance between the different aerosol components constituting the aerosol column at any given location and time) [62].

The aim of this work is to contribute towards achieving the following scientific objectives:

- Deriving a 15-year climatology of total and dust aerosols and cloud-free surface solar radiation in the Mediterranean Basin.
- Quantifying the long-term impact of total and dust aerosols on the solar radiation/energy in the Mediterranean with respect to an aerosol-free atmosphere.
- Investigating the advantages and limitations of existing model and satellite-based aerosol time series for solar-energy-related applications.

After the description of the datasets and the methodology in Section 2, the AOD and DOD climatology is discussed in Section 3.1 along with the intercomparison between MIDAS and CAMS datasets. Section 3.2 presents the aerosol and dust effects on the surface GHI and DNI based on both datasets. In Section 3.3, focus is given on the radiative effects for each DSSI component under extreme dust outbreaks. The interannual variability and trends of the effects of aerosols and dust on GHI and DNI components are presented in Section 3.4, and the clear-sky climatology of GHI and DNI is given in Section 3.5. Finally, our concluding remarks are provided in Section 4.

## 2. Data and Methodology

In the present study, the aerosol and especially dust effects on DSSI in terms of GHI and DNI were investigated over the Mediterranean Basin. Initially, two different AOD and DOD datasets were explored, the newly developed satellite-based MIDAS and the model-derived CAMS reanalysis datasets. As a second step, the quantification of the long-term effects of total aerosols and dust on different DSSI components with respect to aerosol-free conditions were derived, using both AOD and DOD datasets as inputs to radiative transfer model (RTM) simulations. For the simulations, additional aerosol optical properties and atmospheric parameters were used as inputs as well. An intercomparison of the results from the two datasets was performed in order to address the last two scientific objectives listed in Section 1. The description of the utilized datasets and the RTM simulations are provided in Sections 2.1 and 2.2, respectively.

### 2.1. Data

We have used various aerosol and atmospheric-related parameters as inputs in the RTM. Table 1 presents an overview of the used datasets, and a more analytical description is provided in the corresponding subsections.

**Table 1.** Datasets of total aerosol and dust optical properties and key atmospheric parameters for radiative transfer model (RTM) simulations of downwelling surface solar irradiance (DSSI).

| Parameter | Description (Spatial–Temporal Resolution) | Source | Reference |
|---|---|---|---|
| Aerosol optical properties | Satellite-retrieved aerosol optical depth (AOD) ($0.1° \times 0.1°$, 1 day) | ModIs Dust Aerosol (MIDAS) | [55] |
| | Modeled AOD ($0.4° \times 0.4°$, 3 h) | Copernicus Atmospheric Monitoring Service (CAMS) reanalysis | [61] |
| | Single scattering albedo (SSA) ($1° \times 1°$, 1 month) | Max-Planck Aerosol Climatology (MACv2) | [63] |
| | Ångström exponent (AE) ($1° \times 1°$, 1 month) | MACv2 | [63] |
| Dust optical properties | Satellite-based dust optical depth (DOD) ($0.1° \times 0.1°$, 1 day) | MIDAS | [55] |
| | Modeled DOD ($0.4° \times 0.4°$, 3 h) | CAMS reanalysis | [61] |
| | Dust SSA (DU SSA) ($1° \times 1°$, 12 monthly means) | MACv2 | [63] |

**Table 1.** *Cont.*

| Parameter | Description (Spatial–Temporal Resolution) | Source | Reference |
|---|---|---|---|
| Water vapor | Modeled total column water vapor (TCWV) (0.4° × 0.4°, 3 h) | CAMS reanalysis | [61] |
| Ozone | Satellite-retrieved total ozone column (TOC) (1° × 1°/1° × 1.25°, 1 day) | Ozone Monitoring Instrument (OMI) TOMS-Like Level 3 product/Earth Probe (EP) Total Ozone Mapping Spectrometer (TOMS) Level 3 version 8 product | [64,65] |

### 2.1.1. AOD and DOD—ModIs Dust Aerosol (MIDAS) Dataset

MIDAS [55] constitutes a global fine-resolution (0.1° × 0.1°) dataset providing columnar dust optical depth (DOD) at 550 nm, on a daily basis, over a 15-year period (2003–2017). In brief, MIDAS DOD product has been developed through the synergy of quality assured MODIS-Aqua Level 2 AOD and the dust fraction (MDF) to the total aerosol load, in optical terms, acquired from the MERRA-2 reanalysis. Along with DODs, it also provides the associated grid-cell uncertainty estimated using reference AERONET retrievals [66] and LIVAS [67] products for AOD and MDF, respectively. A comprehensive evaluation of the MIDAS DOD versus AERONET DOD-like and an intercomparison against MERRA-2 and LIVAS DODs justified its reliability and validity as well as its caveats which should be taken into account. For the estimation of the radiative effects attributed to the total aerosol load, we used the MODIS-Aqua AOD stored in the MIDAS files. Actually, the MIDAS AOD is the raw MODIS-Aqua AOD (Collection 6.1; [68]), produced by merging Dark Target and Deep Blue retrievals according to Sayer et al. [69], on which quality filters (see Section 2.1 in [55]) have been applied, and it has been reprojected on an equal latitude–longitude grid to that of DOD.

### 2.1.2. AOD, DOD and Total Column Water Vapor (TCWV)—Copernicus Atmospheric Monitoring Service (CAMS) Reanalysis Dataset

The CAMS reanalysis, available from 2003 onwards, is the global reanalysis dataset of atmospheric composition of the European Centre for Medium-Range Weather Forecasts (ECMWF), consisting of three-dimensional time-consistent atmospheric composition fields, including aerosols and chemical species [61]. It is based on ECMWF's Integrated Forecast System (IFS), including an aerosol module described in Morcrette et al. [70]. Five species of tropospheric aerosols are included in the CAMS aerosol model, including dust. For dust sources, the parameterization of Ginoux et al. [71] is implemented. The satellite-derived aerosol products that were assimilated in the CAMS reanalysis were the MODIS-Aqua and MODIS-Terra AOD retrievals [58] and, in addition, the retrievals from the Advanced Along-Track Scanning Radiometer (AATSR) onboard Envisat from 2003 to March 2012. More details regarding the updates in the meteorological part of IFS and in the aerosol and chemical modules, the data assimilation process and the emission datasets are given in Innes et al. [61] and the references therein. CAMS reanalysis products are available from the Copernicus Atmosphere Data Store (ADS, https://ads.atmosphere.copernicus.eu/#!/home (accessed on 25 January 2022)) on a 3-hourly basis. AOD and DOD at 550 nm and TCWV were obtained programmatically for the same period with the MIDAS dataset (2003–2017) on a 0.4° × 0.4° lat/lon grid.

### 2.1.3. Additional Aerosol and Dust Optical Properties (SSA and AE)

For the additional UV–visible–near-IR optical properties for all aerosols and dust, climatological values from the second version of the Max-Planck Aerosol Climatology (MACv2) [63] were utilized, which are available at a global scale with a 1° × 1° spatial resolution. The interannual variability of total aerosol optical properties is provided over

the time period 2001–2016 in terms of monthly means. Monthly climatological values corresponding to the same period are provided for five aerosol species, including dust.

In the current analysis, SSA at 550 nm was used for total aerosols and dust (DU SSA). AODs at 470 nm and 850 nm, for total aerosols, were obtained from MACv2 and were used to calculate the Ångström exponent (AE) (AE 470–850 nm). For dust, a fixed climatological value of 0.4 for AE 440–675 nm was used as proposed by Taylor et al. [72] for the region of study.

### 2.1.4. Total Ozone Column (TOC)

To obtain TOC data for the entire period, a new dataset was constructed by combining data from Ozone Monitoring Instrument (OMI) onboard NASA's Aura satellite from 1 October 2004 until 31 December 2017 and from Total Ozone Mapping Spectrometer (TOMS) onboard the Earth Probe (EP) satellite from 1 January 2003 to 30 September 2004. The satellite-based TOC retrievals were collected from the daily global OMI TOMS-Like TOC Level 3 (OMTO3d) gridded on a $1° \times 1°$ grid product [64] and from the EP TOMS Level 3 (TOMSEPL3) version 8 product [65], which provides daily data on a global grid of $1° \times 1.25°$.

### 2.2. Methodology

### 2.2.1. Spatial and Temporal Extent of the Study

The study was performed for the domain that is confined between 27°N–50°N and 15°W–40°E, which includes the counties around the Mediterranean Sea, as well as part of Central Europe and the Middle East. Analysis was performed with a spatial resolution of $0.4° \times 0.4°$ and for the period 2003–2017.

### 2.2.2. Library for Radiative Transfer (libRadtran) Simulations

The simulations of DSSI components for cloud-free conditions were performed using the uvspec model from the libRadtran package [73]. Using the radiative transfer solver sdisort [74], pseudospectral simulations were performed with a resolution of 1 nm for the spectrum range of 280–3000 nm, using for the molecular absorption the parameterization of LOWTRAN band model [75], as adopted from the SBDART code [76]. The Kurucz 1.0 nm [77] extraterrestrial solar spectrum and the standard US atmospheric profile [78] were utilized, and the surface albedo was set to 0.2.

### 2.2.3. Database (DB) for Radiative Properties

For quantifying the impact of total aerosols and dust on the DSSI components (GHI and DNI), we have performed RTM simulations (see Section 2.2.4) using satellite (MIDAS) retrievals and reanalysis (CAMS) products of AOD and DOD as inputs, complemented by additional aerosol optical properties and atmospheric parameters acquired from the MACv2 climatology, spaceborne observations (OMI, TOMS) and reanalysis products (CAMS), which are described in detail in Section 2.1.

The above datasets differ in spatial and temporal resolution. Initially, the spatial and temporal homogenization of datasets with missing values was performed, which was then followed by the geolocation and synchronization among all datasets in order to generate a complete database (DB) of all the input parameters needed for the RTM simulations (SZA, AOD or DOD, SSA, AE, TCWV, TOC) on a $0.4° \times 0.4°$ lon/lat grid, on an hourly basis, which was selected to be the frequency of RTM simulations in order to account for the sun elevation. Figure 1 provides a schematic overview of this process.

The MIDAS dataset was aggregated to the coarser spatial resolution of $0.4° \times 0.4°$ in order to achieve a small number of missing values. The median value was selected as the aggregation method, as a nonparametric measure of central tendency, based on the findings of Sayer and Knobelspiesse [79]. In order to homogenize the MIDAS dataset in time and space, the missing values were filled by monthly means. Seasonal means were utilized when the monthly data availability was low (<20%). In cases with low seasonal availability,

the spatial gaps were filled using bilinear interpolation. For the 1 h RTM simulations, daily MIDAS values were used and the AOD and DOD were assumed invariant in the day.

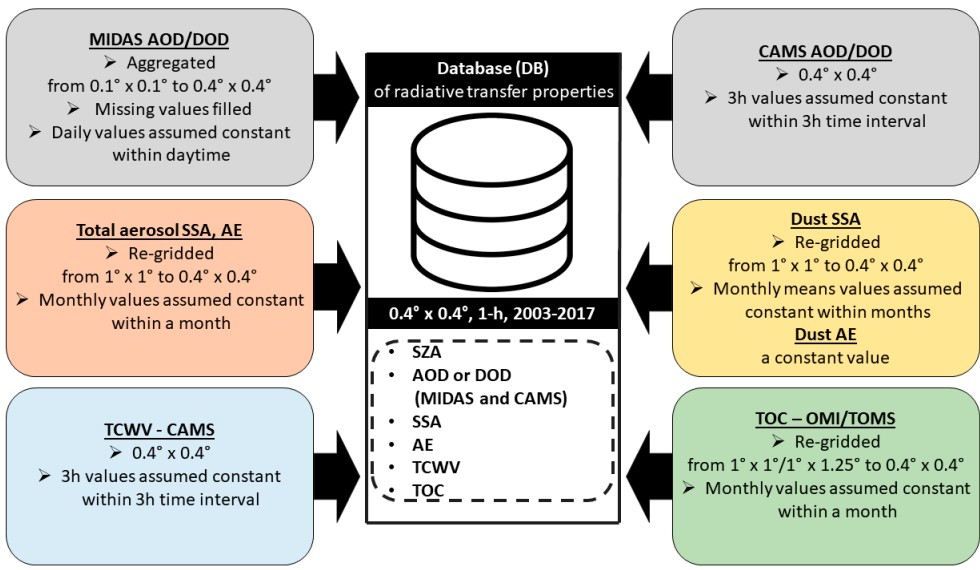

**Figure 1.** Schematic overview of the database (DB) created for the RTM simulations, on a 0.4° × 0.4° lon/lat grid and 1 h temporal resolution.

The diurnal variability of CAMS datasets (AOD, DOD and TCWV) was taken into account and the 3 h values were assumed invariant within each 3-h time interval. For TOC, a temporal fitting was performed filling the days missing with monthly mean values, and a spatial bilinear fit was performed in order to fill the gaps in space. Both OMI and TOMS datasets were bilinearly interpolated to a 0.4° × 0.4° grid. Again, TOC was assumed invariant in the day. The 1° × 1° fields of MACv2 were also bilinearly interpolated to the 0.4° × 0.4° grid and the monthly values of SSA and AE were used, assuming that they remain constant during each month.

### 2.2.4. RTM Methodology

The hourly values of clear-sky DSSI in terms of global and direct components were obtained relying on precalculated look-up tables (LUTs), in order to achieve realistic computational times, as similarly done in Kosmopoulos et al. [80]. Spectral LUTs were constructed containing simulated surface spectral irradiances (global and direct) for a wide range of SZAs and atmospheric factors affecting DSSI (Figure 2a) under cloudless conditions. The simulations for the creation of the LUT were performed using uvspec (Section 2.2.2). All possible combinations of the input parameters of Figure 2a resulted in 74,520 libRadtran simulations of the spectral LUTs. The output spectral irradiances were integrated over the whole SW spectrum to obtain the total irradiances. In order to discretize further, we applied interpolation on the spectrally integrated values, and finer LUTs (Figure 2b) were derived, as they could result from over 200 million hypothetical RTM runs.

Using the fine LUTs and inputs from DB described in Section 2.2.3, instantaneous values of total irradiances (global and direct) every 1 h during daytime were extracted for each grid cell (0.4° × 0.4°) for 2003–2017. This procedure was repeated five times, as different experiments, described in Table 2, in order to quantify the total aerosol as well as the dust effect on different DSSI components, for the two different datasets (MIDAS, CAMS).

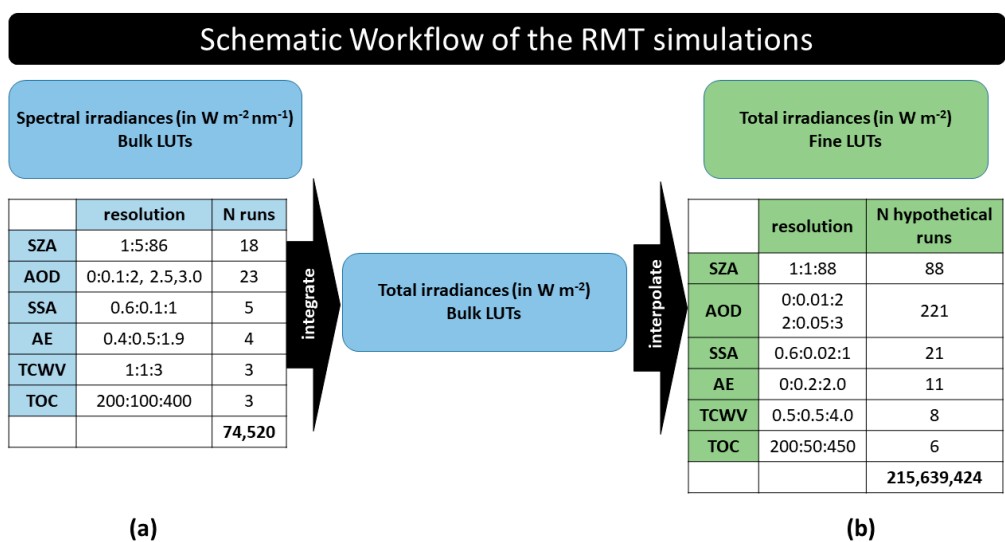

**Figure 2.** Schematic workflow of the RTM simulations. (**a**) The resolution of the input parameters used for simulating spectral global and direct irradiances using uvspec. (**b**) The final resolution of the fine look-up tables (LUTs) of clear-sky total global and direct irradiances after interpolating to all dimensions.

**Table 2.** The different experiments for which DSSI values were extracted from the fine LUTs.

| Atmospheric Conditions | Dataset |
|---|---|
| Aerosol included | MIDAS AOD |
| | CAMS AOD |
| Dust included | MIDAS DOD |
| | CAMS DOD |
| Aerosol-free | AOD = 0 |

Integrating the 1 h instantaneous values of total irradiances over sunlight hours, the daily global horizontal irradiation (GHI) and direct normal irradiation (DNI) components (in $MJ/m^2$) were calculated, and those values were post-corrected for the Earth–Sun distance, and for the surface elevation following the methodology described in Fountoulakis et al. [47]. Using LUT instead of simulating DSSI for the exact conditions of each time step induced some additional uncertainty in the results, which however is small. By comparing the daily integrals using both approaches for particular grid points, it was estimated that the additional (2-fold) uncertainty in the daily integrals is ~0.2 $kW/m^2$, which for the domain of study corresponds to less than 1% of the simulated DSSI in spring, summer and autumn and to less than 10% at the northernmost latitudes of the domain in winter. The corresponding uncertainty in the monthly integrals is much smaller, less than 1% in all cases.

Using the simulated daily irradiations, mean annual and seasonal integrals (INTs) of GHI and DNI, for cloudless conditions, were calculated for the five different experiments described in Table 2 using the following formula:

$$((INT_{i\_on} - INT_{aer\_free})/INT_{aer\_free}) \times 100\% \tag{1}$$

where i stands for aerosols and dust. The relative change (expressed in %) in DSSI due to aerosols and dust presence was calculated with respect to an aerosol-free atmosphere. Using the same formula, the daily relative changes were calculated as well.

2.2.5. Extreme Dust Events

The broader Mediterranean Basin is an area frequently affected by dust outbreaks [57,81], resulting in extremely high concentrations of mineral particles and maximized DODs. Thus, we aimed to quantify the impact of those extreme DOD values on GHI and DNI. To this end, the methodology proposed by Gkikas et al. [82] was applied to MIDAS DOD values in order to define the extreme dust episode days (eDEDs) by also adapting the objective and dynamic algorithm of Gkikas et al. [83] to MIDAS DODs. First, for every pixel, the mean DOD ($\overline{\text{DOD}}$) and the associated standard deviation ($\sigma_{\text{DOD}}$) values were calculated, using the daily values of DOD over the time period 2003–2017. An extreme dust episode occurs on a specific day and at a specific location (pixel) when DOD values are higher than a critical value (threshold):

$$\text{DOD} \geq \overline{\text{DOD}} + 4\sigma_{\text{DOD}} \tag{2}$$

This algorithm is characterized as dynamic since the DOD threshold values are not constant for each pixel. Finally, in order to define a day as an eDED, at least 300 pixels should undergo an extreme dust episode, providing that the data availability for this day is more than 50%. From our analysis, 67 eDEDs were found for the whole study period 2003–2017, or on average 4.5 eDEDs year$^{-1}$.

## 3. Results and Discussion

### 3.1. Satellite-Derived and Modeled AODs and DODs—Climatology and Intercomparison

One of the aims of this study is to investigate the benefits and the drawbacks of choosing between satellite- and model-derived AOD and DOD datasets for estimating their radiative effects. To this end, as described in Section 2, MIDAS and CAMS datasets were explored. For the MIDAS–CAMS comparison, the aggregated MIDAS datasets were used, before filling in the missing values (see Section 2.2.3). CAMS datasets, which have a diurnal variation (3 h time resolution), were synchronized with MIDAS datasets (MODIS-Aqua overpass time) in order to achieve an exact collocation. In Figure 3, the MIDAS cloudless sky data availability (expressed in percentage) is illustrated. Three regions can be distinguished. North Africa, which, due to its scarce cloudiness, has the highest data availability, with more than 70% of daily satellite retrievals with respect to the whole period. Over the Mediterranean Sea, data availability decreases down to 60%. Over Europe, the MIDAS data amount further decreases and is minimized (~20%) in mountainous regions (i.e., Alps). For the temporal aggregation, only grid points with at least 20% data availability on annual and seasonal bases were used, to ensure the representativeness of the results.

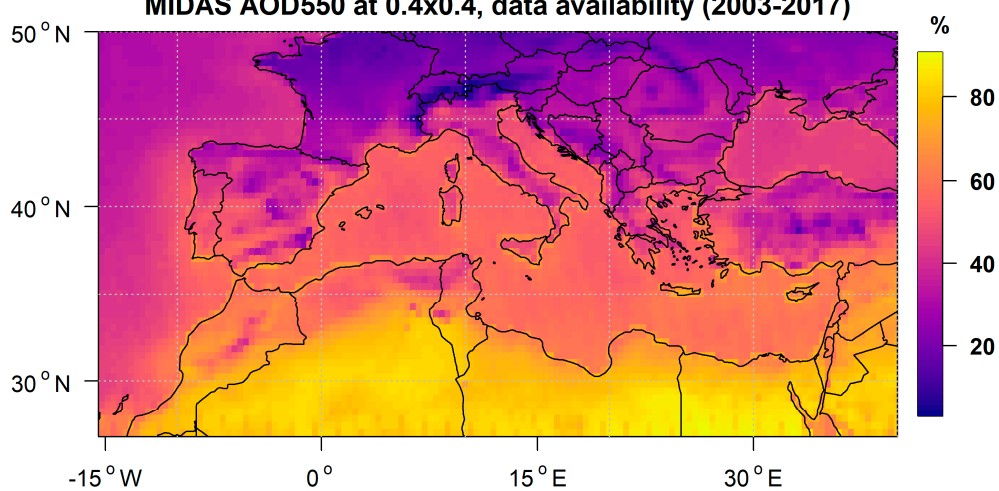

**Figure 3.** MIDAS dataset data availability.

The geographical distribution of the long-term annual averaged values of AOD (Figure 4a,b) and DOD (Figure 5a,b) were derived for both datasets. The corresponding seasonal results are given in the Supplementary Materials (Figures S1–S4). Our analysis expands further CAMS DOD product evaluation [84] in terms of its spatial and temporal variability performance. The frequency histogram of the CAMS–MODIS biases and their mean annual geographical distribution are presented in panels c and d, respectively, for AOD (Figure 4) and DOD (Figure 5). By performing *t*-test for the differences shown in Figures 4d and 5d, the majority of differences were found to be statistically significant (not shown in the figures for clarity) on a 95% confidence level (*p*-value < 0.05).

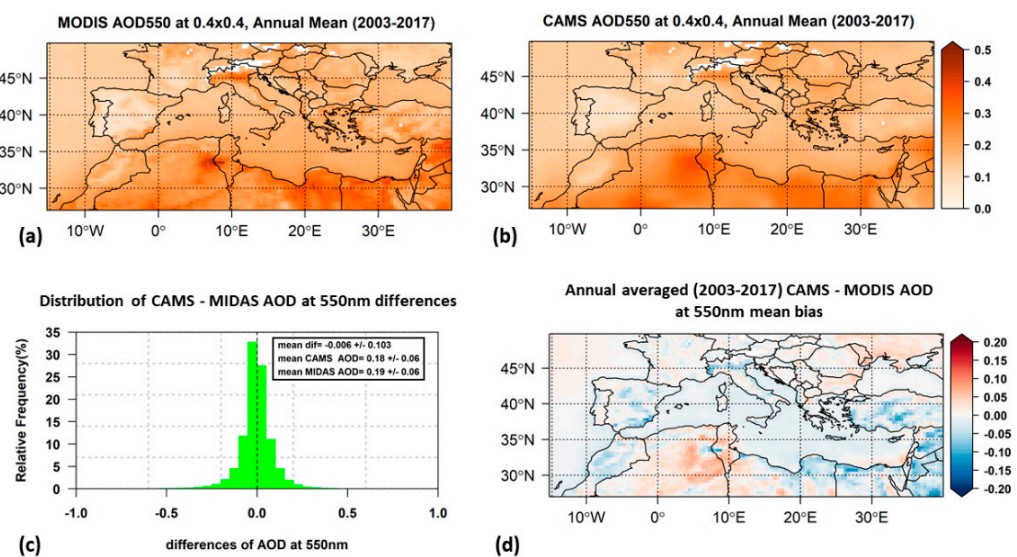

**Figure 4.** Geographical distribution of long-term average of annual mean AOD at 550 nm from MODIS (**a**) and CAMS (**b**). Frequency distribution of CAMS–MODIS AOD biases with their mean value (**c**) and geographical distribution of annual mean biases (**d**). Blank grid points are those that did not fulfill the criterion of at least 20% data availability on annual basis.

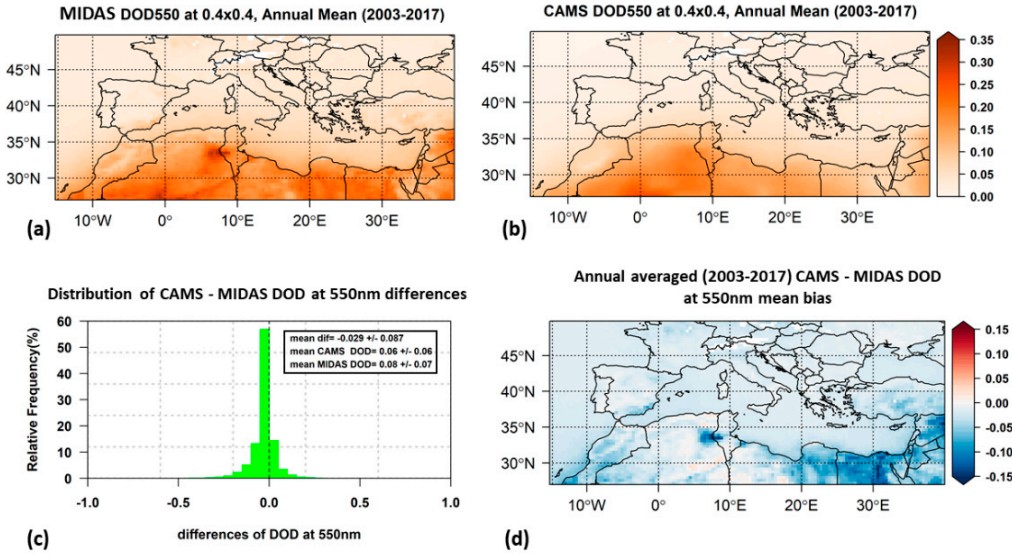

**Figure 5.** Geographical distribution of long-term average of annual mean DOD at 550 nm from MIDAS (**a**) and CAMS (**b**). Frequency distribution of CAMS–MIDAS DOD biases with their mean value (**c**) and geographical distribution of annual mean biases (**d**). Blank grid points are those that did not fulfill the criterion of at least 20% data availability on annual basis.

### 3.1.1. Aerosol Optical Depth

In general, the AOD spatial features are similarly reproduced for both datasets (Figure 4a,b), and the regional averages are almost equal (0.19 ± 0.06 for MODIS and 0.18 ± 0.06 for CAMS, Table 3). Our findings are also in good agreement with those of previous studies focusing on the same region [30,33,85]. Between the two datasets, differences were found in the magnitude of the maximum AOD levels. The annual mean AOD values for each individual pixel range from 0.05 to 0.48 for MIDAS and from 0.05 to 0.37 for CAMS. For both datasets, the maximum and minimum AOD values were found over the same areas. Over North Africa and parts of the Middle East, maximum AOD values were derived that were mainly attributed to desert dust. Large AOD values related to anthropogenic activity [86,87] were found over the megacity of Cairo, Egypt, and the Po Valley, Italy. Low AOD values (0.05–0.15) were found over most of the Iberian Peninsula and southern France, which is in agreement with Obregon et al. [88], who attributed the low AOD to the clean air masses that were transferred over these areas from the Atlantic Ocean due to the westerly air flow (Obregon et al. [88] and the references therein).

Overall, the CAMS-simulated AODs are slightly underestimated (mean bias −0.006) with respect to MIDAS AOD (Figure 4c), which was found to be statistically significant on the 95% confidence level ($p$-value < 0.05, $t$-test for the differences). This is in agreement with the results of previous studies [84], where lower CAMS AODs relative to MODIS were reported over the NAMEE domain. The geographical distribution of the annual mean bias (Figure 4d) revealed areas with annual mean bias that differed a lot from the average value for the whole region. The most significant negative differences were found over an extensive area of Northeast Africa and parts of the Middle East (with annual mean bias up to −0.14), which can be explained by the AOD overestimation of MODIS Dark Target and Deep Blue combined product over these areas [68]. There are also areas with much higher CAMS AOD values relative to MIDAS. Maximum positive differences (up to 0.1) were found over Northwest Africa, which can be explained by the CAMS model overestimation of the organic matter over that area [84]. This CAMS AOD overestimation is more pronounced in summer (Figure S5c).

**Table 3.** Regional averages of mean annual and seasonal AOD and DOD at 550 nm from MIDAS and CAMS and CAMS–MIDAS bias. The maximum of seasonal values is denoted as bold, emphasizing the peak of the seasonal cycle.

| | AOD | | | DOD | | |
|---|---|---|---|---|---|---|
| | **CAMS** | **MIDAS** | **Mean Bias (CAMS–MIDAS)** | **CAMS** | **MIDAS** | **Mean Bias (CAMS–MIDAS)** |
| **Annual** | | | | | | |
| | 0.18 ± 0.06 | 0.19 ± 0.06 | −0.005 ± 0.025 | 0.06 ± 0.06 | 0.08 ± 0.07 | −0.026 ± 0.021 |
| **Seasonal** | | | | | | |
| **winter** | 0.11 ± 0.05 | 0.13 ± 0.06 | −0.019 ± 0.021 | 0.03 ± 0.03 | 0.06 ± 0.05 | −0.030 ± 0.025 |
| **spring** | 0.21 ± 0.07 | 0.21 ± 0.07 | −0.003 ± 0.027 | **0.07 ± 0.07** | **0.11 ± 0.08** | −0.037 ± 0.024 |
| **summer** | **0.23 ± 0.08** | **0.23 ± 0.07** | 0.003 ± 0.038 | **0.08 ± 0.08** | **0.10 ± 0.08** | −0.019 ± 0.027 |
| **autumn** | 0.15 ± 0.06 | 0.16 ± 0.07 | −0.007 ± 0.028 | 0.05 ± 0.05 | 0.07 ± 0.06 | −0.023 ± 0.022 |

There is a clear seasonal cycle (Figures S1 and S2) of the AOD over the Mediterranean Basin. CAMS AOD reproduces the regional patterns of the MODIS AOD seasonal variability quite well, but again there were differences in the magnitude of maximum seasonally averaged AODs between the two datasets. In summer, AODs are maximized (0.66 for MODIS and not exceeding 0.56 for CAMS) over North Africa, particularly in its western parts. Large AOD values were also found over southeast Europe in spring and summer, which are mainly due to emissions of anthropogenic aerosols such as sulfates [33], with a

peak in spring over Po Valley with mean values of 0.37 for MODIS and 0.28 for CAMS. In Table 3, the regionally averaged seasonal mean AOD values are summarized. A distinct seasonal cycle was revealed in both datasets with maximum values during summer and minimum values during winter, which is the same as the seasonal cycle reported by Papadimas et al. [30]. The seasonal variations of AOD are linked to the atmospheric circulation and the meteorological conditions over the study area that are affecting the aerosol emission, removal and transport processes [82,89].

### 3.1.2. Dust Optical Depth

There is a clear latitudinal gradient of DOD (Figure 5a,b). For both datasets, the largest DOD values were found over North Africa and parts of the Middle East, where major dust sources (Sahara and Arabian Peninsula deserts) are located [15,17]. For the annually averaged MIDAS DOD, a maximum of 0.35 was found over a persistently dust-active region of salt lakes (local "chotts") and dry lakes in the borders of Tunisia and northeast Algeria. Large values (0.32) were also found over the desert of central Algeria. Over the eastern Libyan Desert and Egypt, for most pixels, DOD ranges from 0.12 to 0.25. The same range of values was found over the dust sources of Mesopotamia and the Jordan River Basin in the Middle East. A CAMS DOD deficiency is reflected in the systematically lower corresponding values over the aforementioned sources (0.2, 0.26 and 0.05 to 0.15 respectively). Regarding the regional averages (Table 3), there is a small difference between MIDAS and CAMS (0.08 ± 0.07 and 0.06 ± 0.06 respectively) in absolute values.

A relatively high, statistically significant at the 95% confidence level, underestimation (mean bias almost −0.03) of CAMS DODs against MIDAS (Figure 5c) was found. Average CAMS DOD is almost 40% lower compared to MIDAS DOD, which is in agreement with the underestimation of CAMS DOD (up to 46%) with respect to AERONET observations reported by Bennouna et al. [84] over the same area. According to the latter study, the higher CAMS DOD underestimation was found during wintertime, which was attributed to overestimations in biomass-burning organic matter (OM). In summer, the DOD underestimation was attributed to the overestimation of secondary organics over heavily populated areas. The geographical distribution of annual mean bias (Figure 5d) revealed that MIDAS DOD is larger than CAMS DOD almost everywhere. The largest values of CAMS DOD underestimation (up to −0.15) were found over the dust sources of the Saharan and Middle East deserts, not only for the annually averaged DOD but also for the seasonal DOD (Figure S6).

The geographical distribution of DOD over the Mediterranean Basin was found to follow a seasonal cycle (Figures S3 and S4) with maxima in spring and summer, in agreement with the findings of previous studies [33,36,39]. In winter, high DOD levels are confined mainly over northeastern Africa, with DODs up to 0.25 and 0.13 for MIDAS and CAMS, respectively. Dust activity is enhanced in spring, with elevated DOD values (maximum values up to 0.42 for MIDAS and 0.27 for CAMS) over an extended area covering the central and eastern parts of North Africa and the part of the Middle East. In summer, elevated dust levels were found mainly over northwestern Africa, while the highest mean seasonal DOD values were found for this season. In summer, the smallest differences between the two datasets were found (DOD equal to 0.45 for MIDAS and 0.44 for CAMS). This seasonal cycle of dust activity and transport over the Mediterranean is in agreement with previous studies, which also investigated the atmospheric circulation patterns favoring this cycle [41].

The differences that were found between the two different datasets (MIDAS, CAMS), especially regarding the maximum AOD/DOD levels, were investigated further. It was found that a great portion of MIDAS high AOD and DOD values are missing from the CAMS dataset. Table 4 summarizes the amount of data that are higher than 1, 1.5, 2 and 3 in terms of AOD and DOD for both datasets. It also shows the percentage of the missing high values from CAMS datasets compared to MIDAS. For the MIDAS AOD dataset, 0.05% of the values exceed 2, while the corresponding percentage for CAMS is only 0.0015%. For

DOD, over 90% of the missing high values have a lower threshold of DOD 1.5, due to strong CAMS DOD underestimation. It is clear from the results that for very high aerosol burdens there are significant differences between the explored datasets, especially when considering the dust component.

Finally, based on the MIDAS dataset, we estimated that the long-term dust contribution to total aerosols in optical terms ranges from 40% to 90%, over North Africa and the Middle East, making dust the most important aerosol component over these areas.

**Table 4.** Summary statistics of AOD and DOD (from both MIDAS and CAMS datasets) values greater than specific threshold values 1, 1.5, 2 and 3.

|  | **AOD** | | | **DOD** | | |
|---|---|---|---|---|---|---|
|  | **CAMS** | **MIDAS** | **Missing from CAMS Compared to MIDAS** | **CAMS** | **MIDAS** | **Missing from CAMS Compared to MIDAS** |
| **>1** | 0.19% | 0.46% | 57% | 0.08% | 0.31% | 75% |
| **>1.5** | 0.02% | 0.13% | 85% | 0.007% | 0.09% | 92% |
| **>2** | 0.0015% | 0.05% | 97% | 0.0008% | 0.04% | 98% |
| **>3** | 0.000009% | 0.014% | ~100% | 0 | 0.01% | 100% |

### 3.2. Aerosol and Dust Effects on DSSI

In this section, the quantification of total aerosol and dust radiative effects on GHI and DNI over the Mediterranean Basin is presented. Moreover, the effects on DSSI when different datasets (MIDAS or CAMS) of AOD/DOD are used were investigated. For this purpose, in total ~23 million data points were compared. The average number of data points (days) compared for each of the ~8000 pixels of the Mediterranean Basin was ~3000 per pixel or ~200 per pixel per year. As mentioned, missing data are related to MIDAS gaps due to cloudy pixel scenes.

The change in the mean annual integral of GHI due to the presence of total aerosols and dust is presented in Figures 6 and 7, respectively, estimated using MIDAS (panel a) and CAMS (panel b) datasets. The corresponding results for DNI are shown in Figures 8 and 9. The corresponding seasonal results are given in the Supplementary Materials (Figures S7–S14). In all cases, the patterns of GHI and DNI changes are consistent with those of AOD and DOD. The higher the AOD and DOD values are, the higher their radiative effect. Due to the interactions of the incoming solar radiation with the overlying aerosol (dust) layers, the GHI and DNI reaching the surface are reduced with respect to an aerosol-free atmosphere, thus explaining the existence of negative values throughout the domain. The day-to-day variations of these effects are presented in Figure 10. AOD GHI attenuation stands for the GHI reduction by total aerosols and DOD GHI attenuation means the GHI reduction by the dust component. The same nomenclature is used for the DNI component.

### 3.2.1. Aerosol Effects on GHI

There is a qualitative agreement in the geographical distribution of the annual AOD GHI attenuation between the two datasets (Figure 6), and their regional averages using the MIDAS dataset (5.2%, Table 5) are almost the same as those with CAMS (5.1%). The differences in the magnitude of annual mean AODs and especially for the maximum values were also inherited to their radiative effects. The long-term GHI reduction due to aerosols was found to range from 1% to 13% for MIDAS and from 2% to 10% for CAMS.

In general, three subdomains (D) are highlighted for the annual AOD GHI attenuation, based on aerosol load spatial patterns. The highest effects were found over North Africa and the Middle East (D1), where the annual AOD GHI attenuation varies from 4% to 13% based on the MIDAS dataset (4% to 10% for CAMS). Lower values were found for central and southeastern Europe and the Anatolian Peninsula (D2), ranging from 3% to 8% (3% to 7%), with the largest values over the Po Valley. Over the Iberian Peninsula and southern

France (D3), the lowest annual AOD GHI attenuation was found, ranging from 1% to 6% (2% to 5%), with the exception of southeastern Spain (attenuation reaches 8% only for MIDAS dataset). The same low values of the total aerosol effect on the downwelling surface fluxes of the global solar radiation were also found in previous studies [88,90] over the same area.

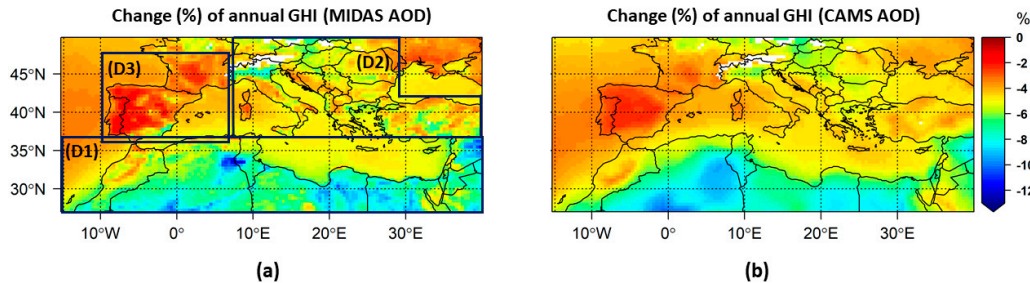

**Figure 6.** Change (in %) of the mean annual integral of GHI due to the presence of aerosols under (**a**) MIDAS AOD and (**b**) CAMS AOD. Blank grid points are those that did not fulfill the criterion of at least 20% data availability on annual basis.

**Table 5.** Change (in %) of the regional averaged mean annual and seasonal integrals of GHI due to total aerosols (AOD) and dust (DOD) from both datasets of MIDAS and CAMS. The maximum of seasonal values is denoted as bold, emphasizing the peak of the seasonal cycle.

|  | AOD | | DOD | |
|---|---|---|---|---|
|  | **CAMS** | **MIDAS** | **CAMS** | **MIDAS** |
|  | **Annual** | | | |
|  | −5.1 (±1.5) | −5.2 (±1.6) | −1.7 (± 1.7) | −2.4 (±1.8) |
|  | **Seasonal** | | | |
| **winter** | −4.9 (±1.5) | **−5.5 (±1.7)** | −1.2 (± 1.2) | −2.3 (±1.7) |
| **spring** | **−5.3 (±1.7)** | −5.3 (±1.8) | **−1.9 (± 1.9)** | **−2.9 (±2.0)** |
| **summer** | −5.2 (±1.9) | −5.0 (±1.7) | **−1.9 (± 2.1)** | **−2.3 (±2.0)** |
| **autumn** | −5.0 (±1.5) | −5.1 (±1.6) | −1.5 (± 1.5) | −2.2 (±1.8) |

There are pronounced seasonal variations (Figures S7 and S8) of the AOD GHI attenuation geographical distribution. The maximum MIDAS AOD GHI attenuation up to 14% (10% for CAMS) was found in spring over North Africa. For CAMS, the maximum reduction was found for summer over northwestern Africa and was 13%, which was similar to the corresponding attenuation for the MODIS dataset. In Table 5, the regional averages of the seasonal AOD GHI attenuation are summarized. The seasonal cycle of AOD GHI attenuation for both datasets differs from the seasonal cycle of the corresponding AOD values. The most notable difference is that the peak of AOD GHI attenuation was derived in winter for the MIDAS dataset instead of summer, when the peak of MIDAS AOD was found. The unexpected peak of the GHI attenuation in winter was mainly due to significant attenuation of the GHI over Egypt and eastern Libya, which was subsequently attributed to minimum seasonal SSA values in winter over the area (Figure S15a). For the CAMS AOD dataset, a small shift of maximum GHI attenuation to spring instead of summer was found.

### 3.2.2. Dust Effects on GHI

Under the absence of nondust aerosol species, the spatial patterns of GHI attenuation, based on MIDAS and CAMS DODs (Figures 7, S9 and S10), show a clear south–north gradient regulated by the reduction in dust load amount from sources to distant downwind regions. The CAMS DOD underestimation is also depicted in the GHI attenuation which is lower than the MIDAS GHI attenuation. Maximum values of the annual DOD GHI

attenuation were found over North Africa and parts of the Middle East, ranging from 2% to 10% for MIDAS and from 2% to 8% for CAMS. This attenuation of GHI by dust accounts for ~45–90% of the overall attenuation by aerosols over this area on annual basis. Dust contribution becomes more significant (up to 95%) on a seasonal basis over the same areas. In summer, the seasonal mean reached 11% for MIDAS and 10.5% for CAMS, over northwestern Africa. Except for summer, the CAMS DOD GHI attenuation is significantly lower than the MIDAS DOD GHI attenuation for the rest of the year.

Regarding the regionally averaged values (Table 5), the annual GHI attenuation due to MIDAS DOD (2.4%) is almost 30% larger than the attenuation estimated for CAMS DOD (1.7%). The seasonal cycle of GHI attenuation attributed to dust is the same as the seasonal cycle of DOD with maxima in spring and summer, with the spring peak being higher by 21% than the summer peak (for MIDAS), which could not be explained solely by the corresponding DOD differences between the two seasons (9%). The sharp MIDAS peak in spring can also be explained by the lower DU SSA values over North Africa and parts of the Middle East during spring (Figure S16b).

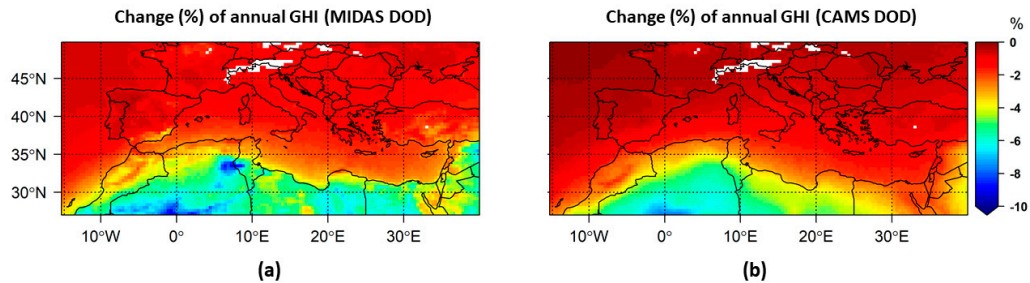

**Figure 7.** Change (in %) of the mean annual integral of GHI due to the presence of dust under (**a**) MIDAS DOD and (**b**) CAMS DOD. Blank grid points are those that did not fulfill the criterion of at least 20% data availability on annual basis.

### 3.2.3. Aerosol Effects on DNI

The average of AOD DNI attenuation (Figure 8) ranged from 5% to 47% for MIDAS and from 10% to 39% for CAMS. The AOD differences between the CAMS and MIDAS datasets were amplified in terms of DNI attenuation. The maximum of DNI attenuation was found for D1 for both datasets, with values ranging from 15% to 47% for MIDAS and to 39% for CAMS. In D1, areas such as Morocco, North Algeria, North Tunisia and the areas around the Red Sea, annual DNI attenuations were less than 20%, which makes these areas favorable for CSP (DNI-related) installations. For D2, an average DNI reduction between 15% and 25% was derived, similar for both datasets, except for Po Valley where CAMS DNI attenuation (26%) was 6% lower than the MIDAS DNI attenuation (32%). The lowest values were found for D3 ranging from 5% to 25% (except for southeastern Spain where it was 35%) for MIDAS and from 10% to 20% for CAMS. The seasonal AOD DNI attenuation values (Figures S11 and S12) reached higher values up to 53% for MIDAS and 49% for CAMS, which were found over Northwest Africa in summer. The seasonal cycle (Table 6) of DNI AOD attenuation followed the corresponding seasonal cycle of the AOD, for both datasets, with maximum in summer and minimum in winter.

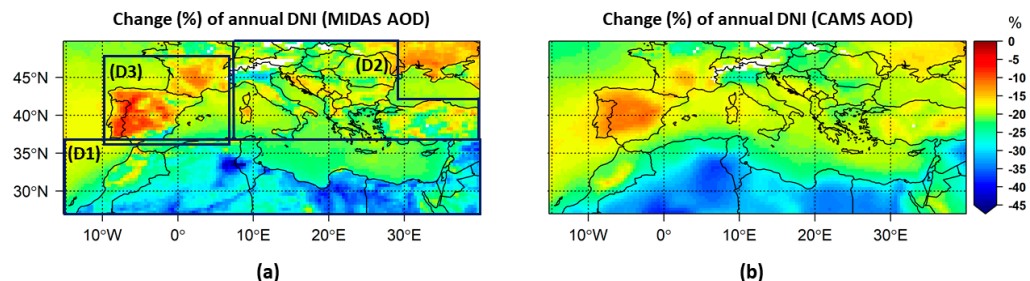

**(a)**

**(b)**

**Figure 8.** Change (in %) of the mean annual integral of DNI due to the presence of aerosols under (**a**) MIDAS AOD and (**b**) CAMS AOD. Blank grid points are those that did not fulfill the criterion of at least 20% data availability on annual basis.

### 3.2.4. Dust Effects on DNI

The peak of DNI attenuation due to dust was found over North Africa and the Middle East, with values ranging from 9% to 37% for MIDAS and from 9% to 28% for CAMS (Figure 9). In summer, over northwestern Africa, the reductions reached values up to 40% and 38% for MIDAS and CAMS (Figures S13 and S14), respectively. The contribution of dust to the overall DNI attenuation by aerosols is ~45–90%. For the regionally averaged values (Table 6), a larger DNI attenuation due to MIDAS DOD (10.7%) was found relative to CAMS DOD attenuation (7.5%). The difference can be attributed to the strong underestimation of CAMS DOD, especially over northeastern Africa and the Middle East.

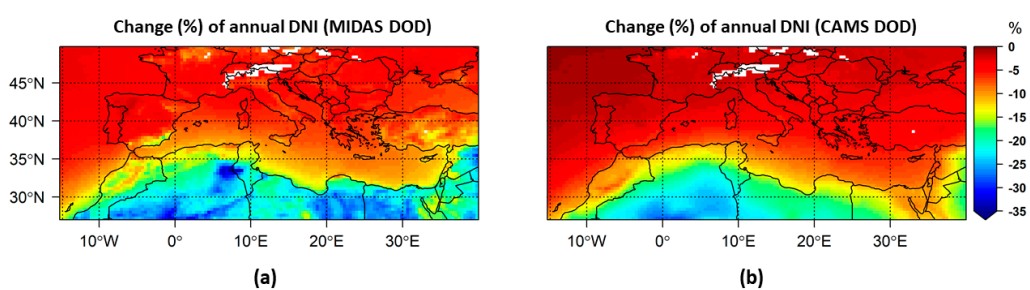

**(a)**

**(b)**

**Figure 9.** Change (in %) of the mean annual integral of DNI due to the presence of dust under (**a**) MIDAS DOD and (**b**) CAMS DOD. Blank grid points are those that did not fulfill the criterion of at least 20% data availability on annual basis.

**Table 6.** Change (in %) of the regional averaged mean annual and seasonal integrals of DNI due to total aerosols (AOD) and dust (DOD) from both datasets of MIDAS and CAMS. The maximum seasonal value is denoted as bold, emphasizing the peak of the seasonal cycle.

| | AOD | | DOD | |
|---|---|---|---|---|
| | **CAMS** | **MIDAS** | **CAMS** | **MIDAS** |
| | **Annual** | | | |
| | −22.2 (±5.6) | −22.1(±5.6) | −7.5 (±6.7) | −10.7 (±7.2) |
| | **Seasonal** | | | |
| **winter** | −17.9 (±5.5) | −19.6 (±6.0) | −4.9 (±4.7) | −9.0 (±7.3) |
| **spring** | −23.6 (±6.2) | −23.2 (±6.7) | **−8.5 (±7.6)** | **−12.7 (±7.9)** |
| **summer** | **−24.0 (±7.3)** | **−23.3 (±6.5)** | **−9.1 (±8.8)** | **−11.2 (±8.2)** |
| **autumn** | −20.5 (±5.8) | −20.5 (±6.2) | −6.4 (±6.1) | −9.4 (±7.2) |

### 3.2.5. Daily Variability

The variability of the daily GHI attenuation due to total aerosols for all pixels (Figure 10a) is much larger than the variability in the annual and seasonal values. The underestimation of CAMS AOD is reflected in the systematically lower values of the daily GHI attenuation. There is no value of GHI reduction due to CAMS AOD above 50%, which is related to the absence of CAMS AOD above 3 (see Section 3.1). It is noteworthy that there are days when aerosols attenuated GHI by ~75% (for the MIDAS AOD). The daily values of GHI reduction due to dust using the MIDAS DOD dataset are constantly larger than those when CAMS DOD is used (Figure 10b), with the only exception being the lower bin around zero. There are days when the MIDAS DOD GHI attenuation exceeded 60%, while the upper limit for CAMS was ~45%. The strong impact of the aerosol particles on the direct component of solar radiation reaching the Earth's surface is depicted in the distributions of the DNI attenuation due to both total aerosols (Figure 10c) and dust (Figure 10d) with values up to 100% for the MIDAS dataset.

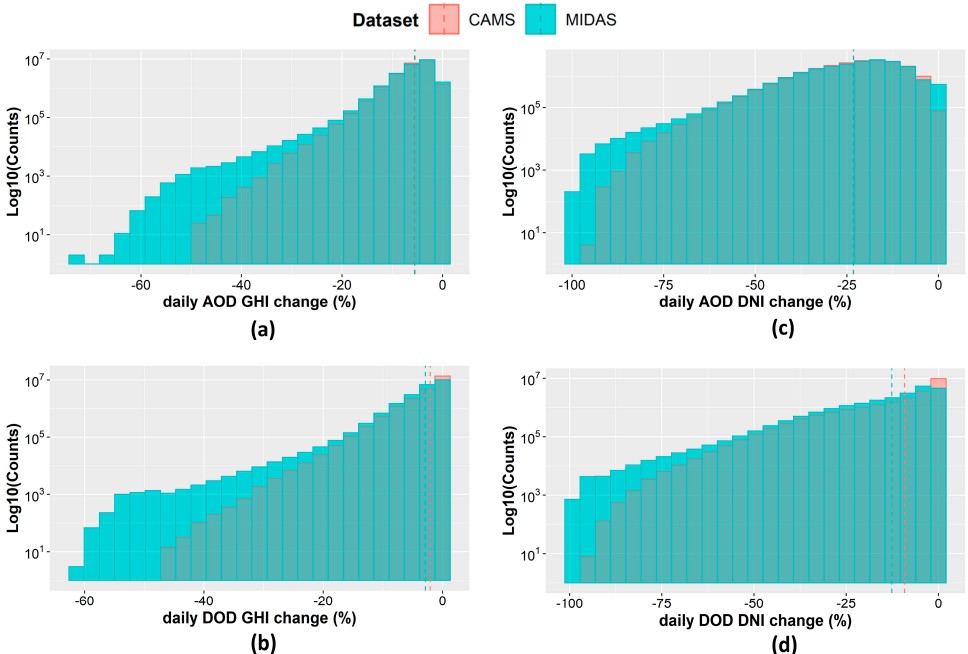

**Figure 10.** Distribution of daily GHI change (%) due to the presence of aerosols (**a**) and dust (**b**) under MIDAS and CAMS AOD and DOD. The same for daily DNI change (%) in panels (**c**,**d**).

The intercomparison between MIDAS and CAMS AOD and DOD effects on DSSI showed how the AOD and DOD differences between the two datasets were expressed in differences in their radiative effects. Underestimation of high AODs and strong DOD underestimation from CAMS were clearly depicted in the attenuation of the DSSI. Thus, the MIDAS dataset is used and discussed in the subsequent analysis. The resulting radiative effects on the surface indicated the important role of total aerosols and especially dust over the Middle East and North Africa (MENA) region, where DSSI attenuation by clouds is comparable or even lower than aerosols.

### 3.3. Extreme Dust Events

Figure 11a presents the geographical distribution of mean MIDAS DOD from the resulting 67 eDEDs. The highest values up to 0.5 of mean DOD from the extremes were found over northwestern Africa, while significantly high mean values up to 0.43 and 0.35 were found also over Egypt and the Middle East, respectively. The highest values of the associated impacts on GHI and DNI, up to 12% and 44%, respectively (Figure 11b,c), were found over northwestern Africa, specifically over southern Tunisia and central Algeria. Large values of eDED mean attenuation were also found over Libya, Egypt and the Middle

East, with values ranging from 4% to 9% for GHI and 17% to 35% for DNI for the bigger part of these areas. Cyprus is the Mediterranean island that was affected the most by the resulting extreme dust events with mean values of GHI and DNI attenuation up to 6.5% and 24%, respectively. For the southern European countries, lower values of mean eDED attenuation were derived, up to 4% for GHI and 19% for DNI, with the exception of very high values over southeastern Spain, up to 5% and 23%, respectively.

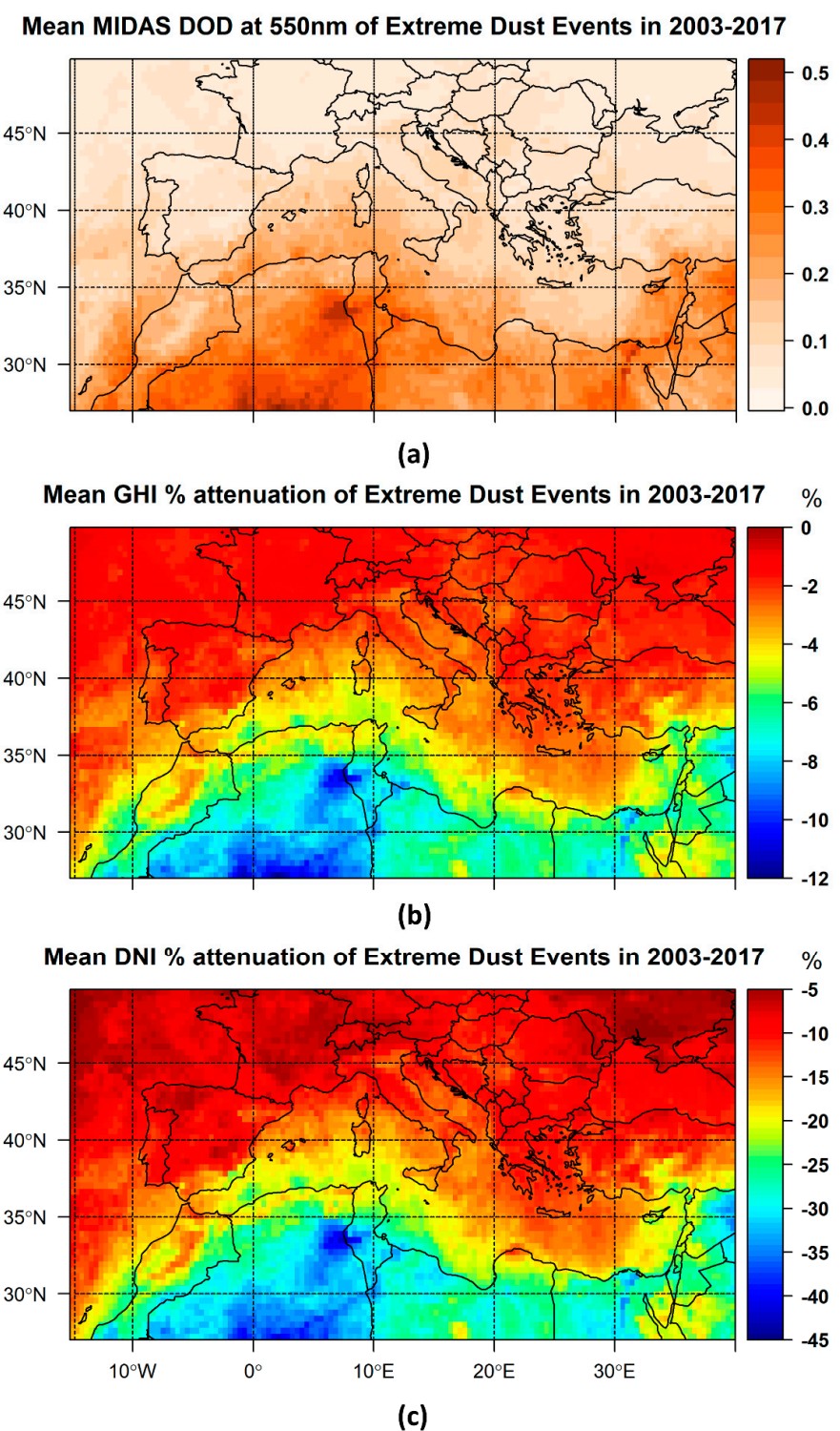

**Figure 11.** Geographical distribution of mean MIDAS DOD at 550 nm for extreme dust episode days (eDEDs) (**a**) and the corresponding GHI (**b**) and DNI (**c**) change (in %).

It should be emphasized here that these results correspond to the long-term (2003–2017) average of the eDEDs and their radiative effects over the region of interest (ROI). Individual dust events are associated with extremely high dust concentrations, resulting in GHI and DNI attenuations up to 50% and 90%, respectively [48].

*3.4. Interannual Variability and Trends*

Using the simulated clear-sky DSSI, we investigated the interannual variability of its changes that were attributed to total aerosols and the dust and nondust components. The nondust optical depth (nDOD), which is considered to be the optical depth of all other aerosol components besides dust, was derived by subtracting DOD from AOD. The main assumption to derive nDOD is that the dust particles are externally mixed with the rest of the aerosol chemical species. For the nDOD RTM simulations, the same additional optical properties as those of total aerosols were assumed. By comparison with the RTM results without aerosols, the corresponding annual GHI and DNI attenuations due to all other aerosol components except dust were derived (nDOD GHI and DNI attenuation hereafter). The interannual variability of the AOD (red line), DOD (blue line) and nDOD (green line) GHI and DNI attenuation is presented (Figure 12b,c) for three different domain averages. The selection of the domains for the spatial averaging was based on the south-to-north gradient of dust, and the geographical limits of those domains are illustrated in Figure 12a.

The year-to-year variability of GHI attenuation by the different aerosol components is weaker (0.5% to 1%) compared to the corresponding variability of the DNI attenuation (2–4%). For the DSSI attenuation (both GHI and DNI) by total aerosols, a successive decline was found after 2008, which is more prominent for D3. This reduction in the AOD DSSI attenuation is in line with the brightening effect over the Mediterranean reported in other studies [91,92]. The resulting decline in the DSSI attenuation by aerosol is attributed mainly to the dust component for D1 and D2, where the variability of annual DOD DSSI attenuation is also large and highly correlated with annual attenuation by total aerosols (correlation coefficients (cc) ranging from 0.85 to 0.92). For D3, which has the sharpest decrease in DSSI attenuation by total aerosols, this is attributed to both dust (cc = 0.84) and nondust components (cc = 0.87).

The increase in GHI (DNI) in D2 and D3 represents the average of positive, statistically significant trends of the order of 1–2% (3–6%)/decade, mainly attributed to decreases in DOD, over the Mediterranean Sea and most of Europe, and negative, nonsignificant, trends over the Anatolian Peninsula. In D1, positive, significant trends in GHI (DNI) of the order of 1% (3–4%)/decade were found over Libya and northwestern Egypt, while negative significant trends of similar magnitude were found over many regions of the remaining D1 area. The overall result was a small positive trend during the whole period (2003–2017), which—as discussed earlier—mainly depicts the increase in 2008–2017. More information regarding the spatial distribution of the trends can be found in the Supplementary Materials (Figure S17).

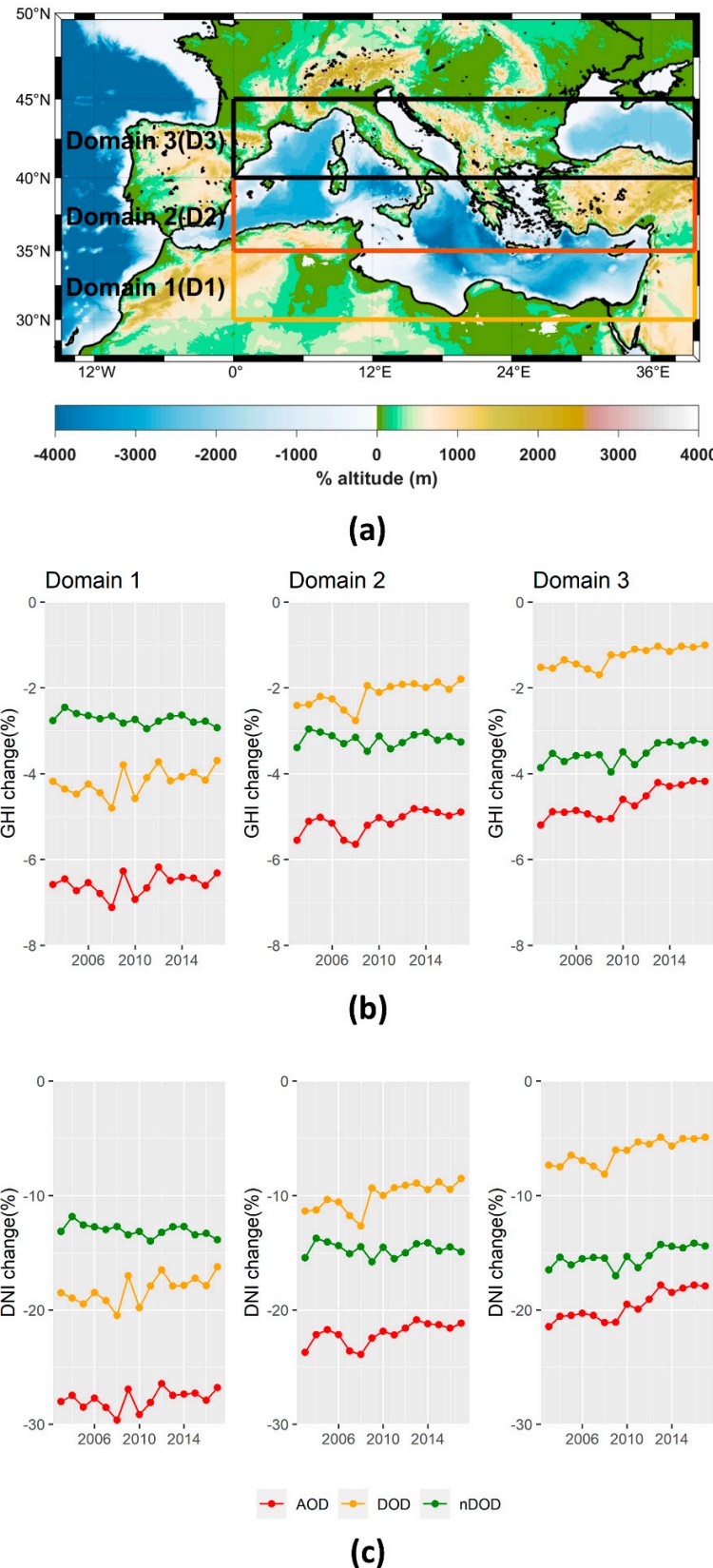

**Figure 12.** (**a**) The geographical limits of the domains used for the spatial averaging. Interannual variability of GHI (**b**) and DNI (**c**) annual integral change (in %) by total aerosols (AOD, red line), dust (DOD, blue line) and other aerosol components beside dust (nDOD, green line) regional averaged for domains 1, 2 and 3 for the 15-year period (2003–2017).

### 3.5. GHI and DNI Clear-Sky Climatology

The availability of solar resources at the Earth's surface is essential information for the different phases of a plant's deployment and operation. Average annual solar irradiation is a primary site selection criterion [44], and a low seasonal variability is preferable in order to match the power demand. According to the results of previous sections, there are areas where the GHI and DNI attenuation due to aerosols can reach 13% and 50%, respectively, which are mainly areas with high solar energy potential (e.g., North Africa). So, the clear-sky GHI and DNI mean annual integrals based on high-quality AOD retrievals are of great importance for such areas with scarce cloudiness.

The clear-sky climatology of GHI and DNI was derived using the MODIS AOD as input in the RTM. Using the daily irradiations (see Section 2.2.4), annual and seasonal integrals of GHI and DNI were derived for every year, and their mean values were calculated for the entire period (2003–2017) and are presented in Figures 13, S18 and S19, while their spatial averages are summarized in Table 7. Given the fact that the cloud effects have not been taken into account, the description of the results is focused on the south part of the domain, over North Africa and the Middle East, which are areas with high solar energy potential, scarce cloudiness and high aerosol loads.

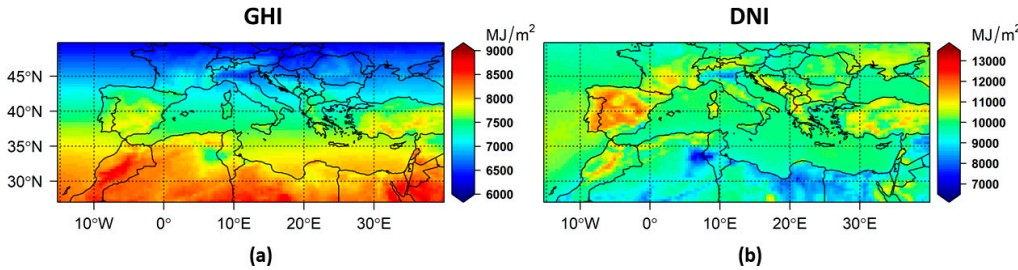

**Figure 13.** Mean (2003–2017) annual integrals for clear-sky GHI (**a**) and DNI (**b**) using MODIS AOD.

**Table 7.** Regional averages of mean annual and seasonal integrals of clear—sky GHI and DNI using ModIs AOD.

|  | GHI (MJ/m$^2$) | DNI (MJ/m$^2$) |
|---|---|---|
| | **Annual** | |
| | 7486 ($\pm$630) | 9899 ($\pm$616) |
| | **Seasonal** | |
| **winter** | 1074 ($\pm$273) | 1972 ($\pm$231) |
| **spring** | 2287 ($\pm$116) | 2747 ($\pm$247) |
| **summer** | 2612 ($\pm$57) | 2931 ($\pm$299) |
| **autumn** | 1512 ($\pm$219) | 2249 ($\pm$143) |

The patterns of both GHI and DNI spatial variability are mainly determined by MODIS AOD (Section 3.1) and the surface altitude. For GHI, a latitudinal gradient (south-to-north) is evident as well. Over North Africa and the Middle East, the cumulative annual GHI and DNI range from 7500 to 8800 MJ/m$^2$ and from 7000 to 12,000 MJ/m$^2$, respectively (Figure 13). Maximum values are observed in the Atlas Mountains (Northwest Africa), in the western parts of Libya and the southeastern parts of the ROI. Regarding the spatial variability of the seasonal integrals, we focused on spring and summer, when the effect of clouds is minimal over the ROI. At this time of the year, the distribution of aerosols expands to the western parts of North Africa, and the spatial variability of DSSI components follows that pattern. While the maximum levels of DNI over the high-altitude areas are ~3300 MJ/m$^2$ in spring and ~3100 MJ/m$^2$ in summer, very low levels of ~1800 MJ/m$^2$ were

found in the same seasons for aerosol-affected areas. These differences are less pronounced for GHI (400 MJ/m$^2$ difference in both seasons).

### 4. Summary and Conclusions

The broader Mediterranean Basin hosts and receives various aerosol types, which are quite variable in spatial and temporal scales. The overarching goal of the present study is to provide an insight into the perturbation of the surface solar radiation, and the subsequent impacts on solar energy production, attributed to the presence of all aerosol types, but with special emphasis on dust. AOD and DOD from two different datasets (MIDAS and CAMS) were used as inputs to the libRadtran RTM (in terms of precalculated LUTs) along with other necessary aerosol and atmospheric parameters under clear-sky conditions. Model outputs were the GHI and DNI, which are of particular interest for different solar power systems (PV and CSP, respectively). Our study domain encloses the broader Mediterranean Basin, and the study period spans from 2003 to 2017 (15 years).

The intercomparison between MIDAS and CAMS datasets revealed that the latter slightly underestimates AOD, and this is mainly evident over areas hosting major aerosol sources, while it strongly underestimates DOD by up to 40% (−0.03) with respect to MIDAS, which is in agreement with the underestimation reported by Bennouna et al. [84] when compared with ground-based retrievals. The CAMS underestimation of high AODs resulted in weaker GHI and DNI attenuations on average by 1–4% and 4–11%, respectively. Likewise, due to the pronounced CAMS DOD underestimation, weaker attenuations were found (by 0.5–4% for GHI and 1–15% for DNI). These findings reveal that using CAMS DOD to describe the radiative effects of dust would give highly uncertain results, especially over areas that are significantly affected by dust, and highlight the importance of using reliable aerosol and dust optical properties to accurately simulate DSSI.

Using the high-quality satellite-derived MIDAS AOD/DOD datasets, a 15-year climatology of total aerosols and dust was established for the broader Mediterranean Basin. The largest AODs were found over dust sources or areas affected by dust transport, with maximum long-term averaged AOD up to 0.48 over Northwest Africa (up to 0.66 for summer season). Over the same area, the peak of MIDAS DOD values was derived as well, with a mean annual value up to 0.35 (up to 0.45 for summer). Dust was found to contribute to total aerosol loads, in optical terms, from 40% to 90% over North Africa and the Middle East, making dust the most important aerosol component over these areas.

Aerosols attenuate GHI by 1–13% and DNI by 5–47%. The largest attenuation (4–13% for GHI and 15–47% for DNI) was found over North Africa and the Middle East. Over the same areas, the GHI and DNI reduction by dust ranged from 2–10% and 9–37%, respectively, contributing by 45–90% to the total aerosol effects on DSSI. During the dry seasons of the year, when the cloud effects over these areas are comparable or even lower than the effects of aerosols, the maximum of aerosol and dust attenuation of the GHI (up to 14% and 11%, respectively) and DNI (up to 53% and 40%, respectively) was found, with the dust being responsible for up to 95% of the AOD DSSI attenuation. On a daily basis, the GHI reduction due to total and dust aerosol reached substantially higher values, up to ~75% and ~60%, respectively. There were days when the DNI component was totally blocked (−100%) under high aerosol and dust loads.

The investigation of the intra-annual variability of the effects of aerosols and dust on GHI revealed, apart from their seasonal variations, the significant role of SSA in calculating the radiative effects of aerosols. The combination of low SSA values with considerable AODs/DODs resulted in peak of regional averaged AOD GHI attenuation in winter, which is reversed compared to the seasonal cycle of MIDAS AOD (maximum in summer and minimum in winter). The same reasons explain the sharp peak of regional averaged DOD GHI attenuation in spring.

The interannual variability of the DSSI attenuation by total aerosol, dust and total aerosol excluding dust was assessed for three subdomains covering the Mediterranean

Basin. After 2008, a successive decline in aerosol effects on DSSI was found for all domains, which was attributed mainly to the reduction in dust.

Since it is well documented that the Mediterranean Basin is frequently affected by dust intrusions, an assessment of the GHI and DNI attenuation was conducted for extreme dust events over the area. Using the MIDAS DOD dataset and by adopting the methodology proposed by [82,83], 67 eDEDs (4.5 eDEDs year$^{-1}$) were identified over the study area for the period 2003–2017. The average DOD during these events reached values up to 0.50 (over North Africa) and the corresponding GHI and DNI attenuations were 12% and 44%, respectively. South Europe was also found to be affected by eDEDs, with the largest GHI and DNI attenuations taking place in southeastern Spain, reaching 5% and 25%, respectively.

Taking advantage of the 15 years of high-quality daily satellite retrievals of AOD from the MIDAS dataset, a clear-sky GHI and DNI climatology for the broader Mediterranean Basin was derived. An added value of this new, clear-sky climatology is that the DSSI values were simulated using, apart from satellite-derived AOD, a climatology of additional aerosol optical properties (SSA, AE), as well as model and satellite products for key atmospheric factors (TCWV and TOC). For the south part of the ROI, in summer when the levels of DSSI are maximum, the main attenuator of GHI and DNI is aerosols (e.g., [47]). Thus, special attention was paid to North Africa and the Middle East, where in summer, high spatial variability of GHI and DNI was found, of 14% and 42%, respectively.

The basic limitations of the study are linked with the RTM inputs and their uncertainties. The main parameter to consider is the optical depth (total aerosol or dust). The uncertainties (MODIS-AOD, MIDAS-DOD, CAMS) have been documented based on the corresponding literature. In addition, SSA and AE data were used as monthly values for a 1° × 1° grid through MACv2 climatology. Day-to-day variability of such parameters can affect the calculated DSSI on a daily basis. However, it has an almost negligible effect when using monthly GHI and DNI for describing the basic climatology of the region under study, especially when optical depth data are relatively accurate. Finally, cloud contamination for satellite-based data is a factor that can affect such DSSI results. However, basic comparison of such data with CAMS modeled data showed no significant systematic optical depth overestimation from the satellite-based data. The same results have been documented when MODIS and MIDAS optical depth comparisons with AERONET have been initiated. Finally, aerosol profiling could have a negligible impact on surface-based calculated irradiances [5].

In conclusion, this study aims to contribute towards a better understanding of the role of aerosols and especially of dust on surface solar radiation in terms of GHI and DNI over the Mediterranean Basin. The results of this analysis, apart from their importance from the perspective of climate science, provide valuable information in terms of management and future planning of PV and CSP installations.

**Supplementary Materials:** The following supporting information can be downloaded at: https://www.mdpi.com/article/10.3390/rs14071535/s1, Figure S1: Geographical distribution of long-term average of seasonal mean AOD at 550 nm from MODIS. Blank grid points are those that did not fulfill the criterion of at least 20% data availability on annual basis, Figure S2: Geographical distribution of long-term average of seasonal mean AOD at 550 nm from CAMS. Blank grid points are those that did not fulfill the criterion of at least 20% data availability on annual basis, Figure S3: Geographical distribution of long-term average of seasonal mean DOD at 550 nm from MIDAS. Blank grid points are those that did not fulfill the criterion of at least 20% data availability on annual basis, Figure S4: Geographical distribution of long-term average of seasonal mean DOD at 550 nm from CAMS. Blank grid points are those that did not fulfill the criterion of at least 20% data availability on annual basis, Figure S5: Geographical distribution of annual mean CAMS–MIDAS AOD biases. Blank grid points are those that did not fulfill the criterion of at least 20% data availability on annual basis, Figure S6: Geographical distribution of annual mean CAMS–MIDAS DOD biases. Blank grid points are those that did not fulfill the criterion of at least 20% data availability on annual basis, Figure S7: Change (in %) of the mean seasonal integral of GHI due to the presence of aerosols under MODIS AOD.

Blank grid points are those that did not fulfill the criterion of at least 20% data availability on annual basis, Figure S8: Change (in %) of the mean seasonal integral of GHI due to the presence of aerosols under CAMS AOD. Blank grid points are those that did not fulfill the criterion of at least 20% data availability on annual basis, Figure S9: Change (in %) of the mean annual integral of GHI due to the presence of dust under MIDAS DOD. Blank grid points are those that did not fulfill the criterion of at least 20% data availability on annual basis, Figure S10: Change (in %) of the mean annual integral of GHI due to the presence of dust under CAMS DOD. Blank grid points are those that did not fulfill the criterion of at least 20% data availability on annual basis, Figure S11: Change (in %) of the mean annual integral of DNI due to the presence of aerosols under MODIS AOD. Blank grid points are those that did not fulfill the criterion of at least 20% data availability on annual basis, Figure S12: Change (in %) of the mean annual integral of DNI due to the presence of aerosols under CAMS AOD. Blank grid points are those that did not fulfill the criterion of at least 20% data availability on annual basis, Figure S13: Change (in %) of the mean annual integral of DNI due to the presence of dust under MIDAS DOD. Blank grid points are those that did not fulfill the criterion of at least 20% data availability on annual basis, Figure S14: Change (in %) of the mean annual integral of DNI due to the presence of dust under CAMS DOD. Blank grid points are those that did not fulfill the criterion of at least 20% data availability on annual basis, Figure S15: Geographical distribution of seasonal mean SSA (MACv2 [63]), Figure S16: Geographical distribution of seasonal mean DU SSA (MACv2 [63]), Figure S17: Trends in % per decade for GHI (panels (a,b)) and DNI (panels (c,d)) due to the changes in AOD (panels (a,c)) and DOD (panels (b,d)).Figure S18: Mean seasonal integrals for clear-sky GHI using MODIS AOD, Figure S19: Mean seasonal integrals for clear-sky DNI using MODIS AOD.

**Author Contributions:** Conceptualization, K.P. and S.K.; methodology, K.P., S.K., I.F., A.G. and P.G.K.; software, K.P. and A.G.; validation, K.P.; formal analysis, K.P. and I.F.; investigation, K.P. and S.K.; resources, A.G. and S.K.; data curation, K.P. and A.G.; writing—original draft preparation, K.P.; writing—review and editing, I.F., A.G., P.G.K., M.H., P.T.N. and S.K.; visualization, K.P.; supervision, S.K., M.H. and P.T.N.; project administration, S.K.; funding acquisition, S.K. All authors have read and agreed to the published version of the manuscript.

**Funding:** This study has been funded by European Commission project EuroGEO e-shape (grant agreement No. 820852).

**Data Availability Statement:** The MIDAS dataset is available at https://doi.org/10.5281/zenodo.4244106 (accessed on 25 January 2022). The CAMS reanalysis datasets were downloaded from CAMS Atmosphere Data Store (ADS) (https://ads.atmosphere.copernicus.eu/cdsapp#!/dataset/cams-global-reanalysis-eac4?tab=overview (accessed on 25 January 2022)). The OMI and TOMS satellite retrievals of TOC were downloaded from https://disc.gsfc.nasa.gov/datasets/ (accessed on 25 January 2022).

**Acknowledgments:** This study has been partly supported (STSM) by COST (European Cooperation in Science and Technology) Action: InDust (CA16202). The MIDAS dataset has been developed in the framework of the DUST-GLASS project (grant No. 749461; European Union's Horizon 2020 Research and Innovation program under the Marie Skłodowska-Curie Actions). This study contains modified Copernicus Atmosphere Monitoring Service information [2020], and neither the European Commission nor ECMWF is responsible for any use that may be made of the Copernicus information or data it contains.

**Conflicts of Interest:** The authors declare no conflict of interest.

## Abbreviations

| | |
|---|---|
| AATSR | Advanced Along-Track Scanning Radiometer |
| ADS | Atmosphere Data Store |
| AE | Ångström exponent |
| AERONET | Aerosol Robotic Network |
| AOD | aerosol optical depth |
| CAMS | Copernicus Atmospheric Monitoring Service |
| CSP | concentrating solar power |
| DB | database |

| | |
|---|---|
| DNI | direct normal irradiation |
| DOD | dust optical depth |
| DSSI | downwelling surface solar irradiance |
| DU SSA | dust single scattering albedo |
| ECMWF | European Centre for Medium-Range Weather Forecasts |
| eDED | extreme dust episode day |
| EP | Earth Probe |
| GHI | global horizontal irradiation |
| IEA | International Energy Agency |
| IFS | Integrated Forecast System |
| INT | integral |
| IPCC AR5 | Intergovernmental Panel on Climate Change 5th Assessment Report |
| IR | infrared |
| LIVAS | Lidar Climatology of Vertical Aerosol Structure |
| LUT | look-up table |
| LW | longwave |
| MACv2 | Max-Planck Aerosol Climatology version 2 |
| MDF | MERRA-2 Dust Fraction |
| MENA | Middle East and North Africa |
| MERRA-2 | Modern-Era Retrospective Analysis for Research and Applications version 2 |
| MIDAS | ModIs Dust Aerosol |
| MODIS | Moderate-Resolution Imaging Spectrometer |
| NAMEE | North Africa, Middle East and Europe |
| nDOD | nondust optical depth |
| OMI | Ozone Monitoring Instrument |
| PV | photovoltaic |
| REari | radiative effects due to aerosol–radiation interactions |
| ROI | region of interest |
| RTM | radiative transfer model |
| SW | shortwave |
| TCWV | total column of water vapor |
| TOC | total ozone column |
| TOMS | Total Ozone Mapping Spectrometer |
| UV | ultraviolet |
| SSA | single scattering albedo |
| SZA | solar zenith angle |

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
