# Peer review of "15-Year Analysis of Direct Effects of Total and Dust Aerosols in Solar Radiation/Energy over the Mediterranean Basin"

_remotesensing, doi:10.3390/rs14071535_

Round 1

Reviewer 1 Report

The goal of this study is to exploit the newly ModIs Dust AeroSol (MIDAS) dataset, quantifying the direct effects on the downwelling surface solar irradiance, induced by the total and dust aerosols amount, under clear sky conditions for the Mediterranean basin. However, major revisions are needed to make before the manuscript is finally accepted for publication.

  • Line 145: CAMS (Copernicus Atmosphere Monitoring Service) reanalysis dataset has low spatial resolution compared to satellite products. Why haven't other data (other satellites or reanalysis) been used to validate the results?

  • Monthly AE and SSA data have been used. Is it not possible to use data with a higher temporal resolution? AE is a basic parameter in the study of aerosols and could be estimated from AOD values at different wavelengths. It would be of great interest if the values of donwelling surface solar irradiance could be estimated with AE values with the same spatial and temporal resolution as AOD.

  • The values of the donwelling surface solar irradiance components have been estimated from pre-calculated Look-Up-Tables without providing the uncertainty of such estimates. It´s more accurate to run the simulations directly, why hasn't this been done? The reduction of the computational cost is not an adequate justification, since the computer that performs these simulations only needs more computing time.

  • The AE, SSA, ozone,... data have a spatial resolution of 1º x 1º. How has it been interpolated to obtain GHI and DNI data at 0.4º x 0.4º? It should be explained.

  • Lines 218-220: “Using the CDS API service, AOD and DOD at 550nm and TCWV were obtained for the same period with MIDAS dataset (2003-2017) on a 0.4º x 0.4º lat/lon grid”. Could you explain in more detail?

  • Lines 372-375: The authors say “The greatest CAMS AOD underestimation (with annual mean bias up to -0.15) was found for the most southeastern part of Spain, an confined area strongly influenced by aerosols, that was also indicated in previous studies with MODIS data [13]”. However, it is in Northeast Africa that the largest underestimates are obtained. Correct this and explain why.

  • Have you validated the MIDAS AOD data with surface AOD measurements, for example from AERONET? It would be advisable for these data to be validated.

  • A remarkable result of this study is the low values of AOD, and consequently the low values of change (in %) of the GHI, in the south of France and in the Iberian Peninsula. This aspect should be studied in more detail. And for this, in addition to the bibliography consulted by the authors [13-16], other articles should also be taken into account, for example (some of them encompass study periods similar to those of this work):
  • Papadimas, C.D.; Hatzianastassiou, N.; Matsoukas, C.; Kanakidou, M.; Mihalopoulos, N.; Vardavas, I. The direct effect of aerosols on solar radiation over the broader Mediterranean basin. Atmos. Chem. Phys. Discuss. 2012, 12, 7165–7185.
  • Floutsi, A.A.; Korras-Carraca, M.B.; Matsoukas, C.; Hatzianastassiou, N.; Biskos, G. Climatology and trends of aerosol optical depth over the Mediterranean basin during the last 12 years (2002–2014) based on Collection 006 MODIS-Aqua data. Sci. Total Environ. 2016, 551–552, 292–303
  • Obregón, M.A.; Costa, M.J.; Silva, A.M.; Serrano, A. Spatial and temporal variation of aerosol and water vapour effects on solar radiation in the Mediterranean Basin during the last two decades. Remote Sens. 2020, 12, 1316.

  • Lines 683-715: the interannual variability by domains is analyzed. However, it would be interesting to calculate the trend of the % change for the entire study area and analyze its spatial variability.

The figures can be improved in the following:        

  • Figure 6: caption should be corrected. “a) CAMS AOD and b) MIDAS AOD” should be replaced by “a) MIDAS AOD and b) CAMS AOD”.

  • Figure 7: caption should be corrected. “a) CAMS DOD and b) MIDAS DOD” should be replaced by “a) MIDAS DOD and b) CAMS DOD”.

  • Figure 8: caption should be corrected. “a) CAMS AOD and b) MIDAS AOD” should be replaced by “a) MIDAS AOD and b) CAMS AOD”.

  • Figure 12 c): “type” should be removed from the legend.

Reviewer 2 Report

The manuscript is presented very well. Authors should also include the trajectory analysis to show that the largest AODs were found over dust sources or areas affected by dust transport.

Reviewer 3 Report

Paper: Remote sensing-1594678. "15-year analysis of direct effects of total and dust aerosols in solar radiation/energy over the Mediterranean Basin".

The aim of this work is to contribute towards addressing the scientific questions below: (i) Derive a 15-year climatology of total, dust aerosols and cloud free surface solar radiation in the Mediterranean basin, (ii) quantify the long-term impact of total and dust aerosols on the solar radiation/energy in the Mediterranean with respect to an aerosol free atmosphere, and (iii) investigate the advantages and limitations of existing model and satellite-based aerosol time series for solar energy related applications.

Comments:

  1. The first paragraph of the introduction is very long. It could be divided into two paragraphs for better understanding.
  2. Remove extra space in L61.
  3. Please give further development to the second paragraph of the introduction. L70-76
  4. There are several very long paragraphs in the introduction. Please adjust.
  5. Synthesize the text inside the parenthesis. L147-149.
  6. I suggest deleting the last paragraph from the introduction. This information would correspond to the chapter on materials and methods.
  7. In the introduction, the main benefits of developing this type of studies are better visualized. Additionally, better visualize its main limitations.
  8. The introduction should be simpler and more concrete.
  9. Please include the websites from which the information used in this study was downloaded.
  10. Please include a nomenclature section on the paper.
  11. Please include a description of the study site. This should be in a conventional chapter of materials and methods.
  12. The paragraph between L292-294 could be integrated with the previous paragraph.
  13. Review the numbering in Table 2.
  14. In the methodology chapter, it is suggested to incorporate more references to support the analyses carried out.
  15. Review the numbering of the figures. Figures 4 and 5 are mentioned in the text first before Figure 3.
  16. It is suggested to synthesize section 3.1. Of the Results and Discussion chapter. Currently extensive and heavy to read. This with is the central axis of the paper: Aerosol and Dust effects.
  17. The information presented between L470-486 should be in the chapter on materials and methods.
  18. Highlight more clearly the quantitative differences between MIDAS and CAMS. Possibly, in a table and for each variable analyzed. Make it clear which of the two methodologies is the validation methodology in this study.
  19. In the introduction make clear the physical differences between aerosols and dust.
  20. The information presented between L646-660 should be in the chapter on materials and methods.
  21. The chapter on results and discussion is very extensive. Synthesize. For example, sometimes this chapter is very descriptive. The main findings should be briefly highlighted.
  22. The conclusions are very extensive. Please synthesize.
  23. Please include the main limitations of this study.
  24. Please include references for 2022.
  25. In general terms, the article should be more synthetic and less descriptive. Please give greater relevance to the findings detected. This is in results and discussion, and conclusions.

Round 2

Reviewer 1 Report

Accept in present form

Reviewer 3 Report

Accept in present form. However, english language and style are fine/minor spell check required.